# Evolutionary Dynamics of Chloroplast Genome and Codon Usage in the Genus *Diospyros* (Ebenaceae)

**DOI:** 10.3390/biology14111568

**Published:** 2025-11-09

**Authors:** Jisi Zhang, Zhuo Li

**Affiliations:** Liaoning Key Laboratory of Development and Utilization for Natural Products Active Molecules, Anshan Normal University, Anshan 114000, China; liz347@nenu.edu.cn

**Keywords:** *Diospyros*, chloroplast genome, evolutionary dynamics, codon usage bias

## Abstract

**Simple Summary:**

*Diospyros* is a large and important genus of trees with ecological and economic value. This study analyzed chloroplast genomes and codon usage in 15 *Diospyros* species to understand their evolution and genetic diversity. The results showed low genetic variation in the IR regions and high conservation at boundary areas, with three main evolutionary groups identified. Codon usage analysis revealed a preference for A or U at the third position and weak codon bias overall. Natural selection, rather than mutation pressure, was found to be the main factor shaping codon usage patterns. These findings provide a robust foundation for future investigations into molecular evolution and phylogenetic relationships in the genus *Diospyros*.

**Abstract:**

*Diospyros*, the most species-rich woody plant genus in Ebenaceae, has attracted significant academic interest due to its ecological and economic importance. This study presented the first complete assembly and annotation of the chloroplast genome of *Diospyros tsangii*. The chloroplast genome measured 157,445 bp, with a typical quadripartite circular structure and 132 annotated coding genes. A comprehensive analysis of evolutionary traits and codon usage preferences across chloroplast genomes of 15 *Diospyros* species were conducted. The main objective was to provide a theoretical basis for understanding phylogenetic relationships and assessing genetic diversity within *Diospyros*. Our findings showed that genetic diversity in the IR regions of the chloroplast genomes is notably lower than that in the LSC and SSC regions. The boundary regions exhibited high conservation with minimal variation. Selected pressure analysis indicated that most coding genes are under purifying selection. Phylogenetic analysis showed that *D. tangii* was sister to *Diospyros oleifera*, and *Diospyros kaki* was closely related to *Diospyros vaccinioides* with high supporting values. The examination of codon usage patterns showed that the GC content at the first, second, and third codon positions of 52 protein-coding sequences followed the order GC1 > GC2 > GC3, with a preference for A or U bases at the third position. The effective number of codons ranged from 45.13 to 45.43, which indicated the weak codon bias. The neutral-plot, ENC-plot, and PR2-plot analysis suggested that natural selection predominantly influences the codon usage patterns in *Diospyros* plants. These results would be vital to understand the evolutionary dynamics of the genus *Diospyros*.

## 1. Introduction

The genus *Diospyros* L. is the largest genus of Ebenaceae and comprises about 485 species of evergreen or deciduous wood plants. This genus widely distributes from tropical to temperate regions all over the world [1]. The highest species diversity is found in the Asia–Pacific region, which hosts around 300 species. Among the 60 documented species of *Diospyros* in China, 45 species are endemic, and 18 species are stenochoric, with a diversity center in southeast and southwest China [2]. As is well known, *Diospyros* is the most economically important genus of the Ebenaceae [3]. Several of these are widely recognized for their valuable timber or edible fruits, while numerous others play a crucial role as sources of medicinal compounds [4,5,6,7,8]. Such as *Diospyros kaki*, which is a globally significant economic fruit tree, known for its nutritional content including tannins, sugars, vitamin C, and carotenoids. Generally, closely related wild species are vital to qualitatively improve the breeding of the persimmon crops. Therefore, it is necessary to accumulate genetic information for exploring the genetic diversity of *Diospyros*.

The chloroplast genome has been proven to be informative and valuable for plant phylogenetic studies [9,10,11]. Typically, it consists of a pair of inverted repeat (IR) regions, a long single-copy (LSC) region, and a short single-copy (SSC) region, containing around 110–130 genes, mainly encoding essential proteins for photosynthesis and energy metabolism [12,13]. Comparative genomic analysis of chloroplast genomes identifies highly variable regions suitable for developing specific molecular markers such as SSRs and DNA barcodes [14,15,16,17]. For instance, Li et al. (2023) effectively resolved the phylogenetic relationships of *Eriocaulon* (Eriocaulaceae) using the complete plastome, and indicated the genus diverged since the late Miocene and diversified in the Quaternary [18]. Similarly, Wang et al. (2023) clarified species relationships within *Lagerstroemia* (Lythraceae) using plastome data and explored a rapid radiation since the late Miocene [19]. Jiang et al. (2023) reconstructed phylogenetic relationships within *Zingiber* (Zingiberaceae) with plastome data and identified 19 genes under positive selection [20]. Furthermore, Huang et al. (2024) provided robust support for classifying of *Engelhardia* species at the sectional level and identified three subfamilies within Juglandaceae, demonstrating the effectiveness of plastome sequences for achieving high phylogenetic resolution [21]. Lastly, Qu et al. (2025) conducted a comprehensive study of chloroplast genome characteristics across diverse cassava genetic resources, significantly enhancing understanding of chloroplast genome evolution in this vital crop species [22]. In general, the chloroplast genomes can effectively resolve phylogenetic relationships at different taxonomical levels.

Additionally, codon usage bias (CUB) refers to the non-random selection of synonymous codons that encode the same amino acid during protein-coding processes, reflecting the adaptability of genetic transcription and translation [23]. CUB is characterized by codon base composition, usage frequency, and evolutionary forces such as natural selection and mutation pressure [24,25]. The evolutionary conservation, lack of recombination, and maternal inheritance of the chloroplast genome make CUB analysis valuable for assessing genetic diversity [26,27,28,29]. For instance, Yang et al. (2024) revealed a consistent preference for codons ending in A or T across 18 *Taraxacum* species, and this observed CUB was mainly influenced by natural selection [30]. While Guo et al. (2025) examined CUB in the chloroplast genomes of ten *Androsace* species and demonstrated that the codons preferred encoding with A or U bases were closer related to mutational pressure than natural selection [31].

In this study, a thorough investigation into the evolution of chloroplast genomes and codon usage bias is conducted to address the genetic diversity within this genus in *Diospyros*. The primary objectives of this study are: (1) comprehensively detecting the chloroplast genomes evolution of *Diospyros*, and (2) exploring codon usage bias among *Diospyros* species and further identifying optimal codons. These findings would be key to the *Diospyros* germplasm resources and enhance our understanding of genome evolution in economical and horticultural plants.

## 2. Materials and Methods

### 2.1. Sampling Collection and Sequencing Procedure

The plant materials of *Diospyros tsangii* were obtained from Jinggangshan in Jiangxi Province, China (26.5818° N, 114.1386° E). The leaves were preserved in silica gel. Total genomic DNA was isolated using a modified CTAB protocol [32]. Following extraction, an Illumina paired-end (PE) library was prepared and sequenced at Personalbio Biotechnology Co., Ltd., located in Shanghai, China. The other *Diospyros* species were downloaded from GenBank (Table 1).

### 2.2. Chloroplast Genome Assembly and Annotation

For chloroplast genome assembly, approximately 6 Gb of raw paired-end reads (150 bp in length) were generated. The Trimmomatic v0.39 software was used to process the raw data by removing adapter sequences and trimming low-quality regions (LEADING:3, TRAINING:3, SLIDING WINDOW:4:15, MINLEN:36), resulting in high-quality clean reads [33]. These filtered reads were then assembled into a complete chloroplast genome using GetOrganelle v1.5 [34]. Annotation of the *D. tsangii* chloroplast genome was performed utilizing GeSeq (https://chlorobox.mpimp-golm.mpg.de/geseq.html (accessed on 2 December 2024)) and Geneious v9.1.4 (http://www.geneious.com/ (accessed on 2 December 2024)), with *D. oleifera* (NC_030787) serving as the reference genome [35]. The fully annotated chloroplast genome sequence of *D. tsangii* has been deposited in GenBank under the accession number PX413321.

### 2.3. Comparative Analysis of Chloroplast Genomes in the Genus Diospyros

#### 2.3.1. Analysis of Simple Sequence Repeat in Chloroplast Genomes

The simple sequence repeat (SSR) characteristics of chloroplast genomes from 15 economically valuable species in the genus *Diospyros* were examined using the MISA tool (https://webblast.ipk-gatersleben.de/misa/ (accessed on 20 February 2025)) [36]. The analysis was conducted with the following minimum repetition criteria: at least 10 repeats for mononucleotides, 6 for dinucleotides, and a minimum of 5 repeats for tri-, tetra-, penta-, and hexanucleotides motifs.

#### 2.3.2. Comparative Assessment of Chloroplast Genome Junction Regions

The IRscope platform (https://irscope.shinyapps.io/irapp/ (accessed on 25 February 2025)) was employed to conduct a comparative examination of the junctions between the large single-copy (LSC), small single-copy (SSC), and inverted repeat (IR) regions, particularly emphasizing the visualization of IR boundary contractions [37]. Furthermore, boundary-associated genes were delineated, and variations in gene types and sizes were scrutinized to evaluate the structural integrity of chloroplast genomes.

#### 2.3.3. Evaluation of Nucleic Acid Polymorphism and Selective Pressure

Nucleotide polymorphism (Pi) was calculated from aligned sequences using DnaSP v6.0 software [38]. A sliding window approach was applied with a window size of 600 bp and a step size of 200 bp, based on the plant chloroplast genome model. Synonymous (*dS*) and non-synonymous (*dN*) substitutions, as well as the *dN*/*dS* ratio, were analyzed by aligning the protein-coding sequences of *D. tsangii* with those of 14 other *Diospyros* species. *D. tsangii* was used as the reference in pairwise gene alignments. The 80 common protein-coding genes were extracted using Geneious Prime 2021, and *dN* and *dS* values were computed using DnaSP v6.0. To assess selection pressures acting on chloroplast genes with distinct functional roles, CDS genes were classified into photosynthesis-related, self-replication-related, and other functional categories (Appendix A). Box plots of *dN*/*dS* values for CDS genes were generated according to functional classifications and taxonomic groups, with significant intergroup differences indicated. All analyses were performed in R version 3.4.4.

#### 2.3.4. Phylogenetic Tree Reconstruction and Time Estimation

Multiple sequence alignments were conducted using MAFFT version 7 [39], with subsequent removal of poorly aligned regions utilizing Gblock version 0.91b [40]. Phylogenetic reconstruction was carried out on complete chloroplast genome sequences using both Bayesian inference (BI) and maximum parsimony (MP) methods. For BI analysis, MrBayes v3.2.6 was employed under the GTR + G substitution model [41]. MP analysis was carried out in PAUP version 4b10 using heuristic search strategies, and nodal support was evaluated through 1000 bootstrap replicates [42].

Divergence times were estimated using BEAST v2.7.7 under a relaxed exponential clock model [43]. The root age was set as 28.01 Ma (S.D. = 2.2) under normal prior [44]. The speciation prior was set as YULE, and the substitution model of DNA regions was set as the GTR +I +G model. Markov Chain Monte Carlo (MCMC) searches were run for 100 million generations and sampled every 5000 generations. Convergence was assessed by Tracer v.1.7 to ensure the effective sampling size (ESS) for all parameters >200 [45]. The maximum clade credibility (MCC) tree was calculated by TreeAnnotator v.2.6.0 [46]. According to Linan et al., *D. ferrea* from Clade B was set as outgroup, and others sampled *Diospyros* species belonged to Clade A [44].

### 2.4. Analysis of Codon Usage Bias Pattern in Chloroplast Genomes in Diospyros

#### 2.4.1. Calculation of Parameters Related to Codon Usage Bias

Biases in calculating preference indices can arise from issues like insufficient sample size and incomplete codon coverage in short genes [47,48]. These biases complicate the differentiation between selection pressure effects and random factors, undermining research credibility. To enhance statistical reliability and biological interpretation accuracy, this study excluded genes shorter than 300 bp. Additionally, gene sequences with non-ATG start codons, abnormal stop codons, and internal stop codons were omitted. A total of 52 protein-coding genes were scrutinized for codon bias across the chloroplast genomes of 15 *Diospyros* species in this investigation.

The software CodonW v1.4.2 was employed to compute the Relative Synonymous Codon Usage (RSCU) and Effective Number of Codons (ENC) [49,50]. RSCU denotes the relative frequency of codon usage for encoding specific amino acids, with values above 1 indicating high preference, values at 1 indicating no preference, and values below 1 indicating weak preference. The data analysis was conducted using IBM SPSS 29.0. Additionally, the overall GC content (GC_all) of each gene’s coding sequence, as well as the GC content at the first (GC1), second (GC2), and third (GC3) nucleotide positions within codons, were determined utilizing the CUSP tool (https://bioinformatics.nl/cgi-bin/emboss/cusp (accessed on 25 February 2025)).

#### 2.4.2. Analysis of the Causes of Codon Usage Bias

To systematically identify the primary forces shaping CUB in *Diospyros*, several quantitative approaches—including neutral-plots, ENC-plots, and PR2-plots—were applied. Initially, the GC contents at the first and second positions (GC1 and GC2) in the coding sequences were computed to derive the average GC12 value. Subsequently, a neutral plot was constructed with GC3 on the horizontal axis and GC12 on the vertical axis to investigate the predominant influence of mutation or selection pressure on CUB. A significant correlation indicates mutation as the primary factor, while an insignificant correlation suggested a greater contribution of selection pressure in shaping codon usage patterns [51,52]. For the ENC-plot analysis, observed ENC values (ENC_obs_) were plotted against GC3s (GC content at synonymous third sites), with expected ENC values (ENC_exp_) calculated using the formula: 2 + GC3s + 29/[GC3s^2^ + (1 − GC3s)^2^]. The ggplot2 package in R was utilized for visualizing the ENC-plot, where discrepancies between ENC_obs_ and ENC_exp_ values reveals the main driver of codon preference [53]. The standard curve represents codon preference solely influenced by mutation in the absence of selective pressure [54]. The PR2-plot method was applied to evaluate the combined effects of mutational pressure and selection on codon usage patterns. This graphical representation utilizes A3/(A3 + T3) on the *y*-axis and G3/(G3 + C3) on the *x*-axis. The position and orientation of each gene on the plot indicates its CUB, with the central point (A = T, C = G) corresponding to balanced codon usage, indicating no bias [55,56].

#### 2.4.3. Determination of Optimal Codons

In the assessment of optimal codons, ENC serves as the benchmark for evaluating codon usage bias. Genes are sorted on their ENC values, with the top 10% and bottom 10% deciles selected to form high- and low-expression gene pools, respectively. Subsequently, the RSCU and the ∆RSCU (calculated as RSCU_high_ − RSCU_low_) for codons in these gene pools are computed. Codons exhibiting both an RSCU greater than 1 and a ∆RSCU of at least 0.08 were classified as optimal codons [49].

## 3. Results

### 3.1. Chloroplast Genome Characters of Diospyros tsangii

The complete chloroplast genome of *D. tsangii*, the newly sequenced species in this study, exhibits a typical quadripartite structure with a length of 157,445 bp, comprising a SSC of 18,523 bp, a LSC of 86,744 bp, and a pair of IRs of 26,089 bp (Appendix A). Annotation revealed a total of 132 genes, including 87 protein-coding genes, 37 transfer RNA (tRNAs) genes, and eight ribosomal RNA (rRNAs) genes (Appendix A). Of these genes, 74 were associated with self-replication, encompassing 11 genes linked to the large ribosomal subunit and 14 to the small ribosomal subunit. Furthermore, 45 genes were implicated in photosynthesis, with 6 genes related to ATP synthase, 12 to NADH dehydrogenase, 6 to the cytochrome b/f complex, 5 to the PS I system, 15 to the PS II system, and 1 associated with Rubisco. Additionally, 13 genes were annotated with either other functions (i*nfA*, *clpP*, *ccsA*, *accD*, *cemA*, and *matK*) or unknown functions (*ycf1*, *ycf2*, *ycf3*, *ycf4*, and *ycf15*). Among the genes, 14 were observed to contain a single intron (*atpF*, *ndhA*, *ndhB*, *petB*, *petD*, *rpl2*, *rpl16*, *rpoC1*, *trnA^UGC^*, *trnG^UCC^*, *trnI^GAU^*, *trnK^UUU^*, *trnL^UAA^* and *trnV^UAC^*), while 3 genes (*rps12*, *clpP* and *ycf3*) harbored two introns (Appendix A).

### 3.2. Comparative Analysis of Chloroplast Genomes in Diospyros

#### 3.2.1. The Size and Structure of the Chloroplast Genome

Based on the newly obtained *D. tsangii* chloroplast genome, we further conducted comparative evolutionary analysis with other fourteen *Diospyros* species (Table 1). Chloroplast genome sizes varied from 157,368 bp (*D. rhombifolia*) to 157,999 bp (*D. hainanensis*), with a consistent GC content of 37.4% across all species. Each chloroplast genome comprised a LSC, a SSC, and two inverted repeat regions (IRa and IRb). The LSC region spanned from 86,774 bp (*D. tsangii*) to 87,523 bp (*D. hainanensis*), accounting for 55.09% to 55.43% of the total genome length. The SSC region ranged from 18,322 bp (*D. hainanensis*) to 18,536 bp (*D. kaki*), representing 11.60% to 11.76% of the genome. The IR regions varied from 25,874 bp (*D. strigosa*) to 26,180 bp (*D. dumetorum*), comprising 32.88% to 33.17% of the total length. The 15 *Diospyros* species had 132 coding genes, including 87 CDSs, 8 rRNAs, and 37 tRNAs.

In total, 48–77 SSR loci were identified within the chloroplast genomes of 15 *Diospyros* plants species (Figure 1A, Appendix A). These loci comprised 29–66 single nucleotide repeats, predominantly A/T base repeats, as well as 2–5 dinucleotide repeats, 1–4 trinucleotide repeats, and 5–10 tetranucleotide repeats (Figure 1B, Appendix A). Furthermore, pentanucleotide repeats were observed in eight species, namely *D. maclurei*, *D. hainanensis*, *D. strigosa*, *D. eriantha*, *D. dumetorum*, *D. rhombifolia*, *D. cathayensis*, and *D. sutchuensis*. Examination of the repeat sequences unveiled the presence of 15–27 forward repeats, 1–6 inverted repeats, and 21–34 palindrome repeats. Additionally, a complementary duplication was identified in *D. maclurei* and *D. dumetorum* (Figure 1C, Appendix A). The results indicated a positive association between the length of the IR and the overall chloroplast genome size, while no significant correlation was found between the number of SSRs and the total chloroplast genome length (Figure 1D,E).

#### 3.2.2. Boundary Analysis of IR in the Genus Diospyros

The collinearity analysis conducted on 15 *Diospyros* species did not reveal any gene rearrangements or inversions (Appendix A). The gene composition near the boundaries of LSC/IRb (JLB), IRb/SSC (JSB), SSC/IRa (JSA), and IRa/LSC (JLA) remained consistent among species, although there were slight variations in the distances between these genes and their respective boundaries (Figure 2). Mapping of the IR boundary revealed that the *rps19* gene in *Diospyros* species spanned from LSC to IRb, with a 7–8 bp extension into the IRb region. Additionally, the *ndhF* gene was located adjacent to the junction between the SSC and IRb (JSB). A 4-bp extension across this boundary was observed in *D. rhombifolia* and *D. cathayensis*, while the remaining species did not exhibit such an extension. Furthermore, the *ycf1* gene was found adjacent to the junction between the LSC and IRa (JSA).

#### 3.2.3. Chloroplast Genome Sequence Variation and Selected Pressure Assessment

Nucleotide diversity in chloroplast genomes of 15 *Diospyros* species was evaluated using DnaSP software, demonstrating an average Pi range from 0 to 0.082 (Figure 3). Notably, regions with high variability were predominantly situated in the LSC and SSC regions, whereas the IR regions displayed greater conservation. Specifically, within the LSC region, the t*rnT*-*trnL* (0.02486), *petA*-*psbJ* (0.02557), and *psbE*-*petL* (0.03183) spacer regions exhibited the highest Pi values. In the SSC region, the *ycf1* (0.082), *rpl32*-*trnL* (0.08065), and *ndhA* (0.08014) three regions showed the greatest diversity.

To assess the evolutionary pressures on protein-coding homologous genes across 15 *Diospyros* species, we calculated the *dN*/*dS* ratios for 80 coding sequences genes (Appendix A, Appendix A). The analysis revealed that 79 of these genes had *dN*/*dS* values below 1, indicating that they are predominantly under purifying selection. Interestingly, the photosynthesis gene *psbI* had a *dN*/*dS* value slightly above 1, suggesting potential positive selection. Furthermore, no significant differences were observed in the *dN*/*dS* values among self-replication-related, photosynthesis-related, and other genes across the 15 *Diospyros* species.

#### 3.2.4. Phylogenetic Analysis

In this study, the BI and MP phylogenetic trees based on complete chloroplast genome sequences of *Diospyros* species were depicted in Figure 4. The phylogenetic relationships among the *Diospyros* have been fully resolved. The *D. sutchuensis*, *D. cathayensis*, and *D. rhombifolia* formed a clade with highest supporting values (BI-PP = 1.00, MP-BS = 100%). Subsequently, *D. dumetorum*, *D. eriantha*, and *D. strigosa* clustered together (BI-PP = 1.00, MP-BS = 100%). Then the two clades clustered with *D. hainanensis* with well supporting values. Additionally, *D. glaucifolia*, *D. lotus*, and *D. morrisiana* formed a clade with highest supporting values (BI-PP = 1.00, MP-BS = 100%). Finally, the four species (*D. tangii*, *D. oleifera*, *D. kaki*, and *D. vaccinioides*) formed a clade with well supporting values (BI-PP = 1.00, MP-BS = 100%). Especially, *D. kaki* was closely related to *D. vaccinioides*.

Our results suggested that *D. tangii* diverged from its sister species *D. oleifera* at 1.62 Ma (95% HPD: 0.18–5.00), and *D. kaki* diverged from *D. vaccinioides* at 0.44 Ma (95% HPD: 0.03–2.57) (Figure 4).

### 3.3. Codon Usage Bias in the Chloroplast Genomes of Diospyros

#### 3.3.1. Codon Composition Characteristics and Preferences of Chloroplast Genomes

Fifty-two conserved genes were identified in CUB analysis of 15 *Diospyros* species (Appendix A). The GC contents of the three codon positions, GC1, GC2, and GC3, ranged from 46.95% to 47.05%, 39.44% to 39.58%, and 27.70% to 27.91%, respectively. The average overall GC content (GC_all) fell within the range of 38.05% to 38.18%. The order of GC content observed was GC1 > GC2 > GC3, indicating a non-uniform distribution of GC content among codon positions. All GC values were below 50%, suggesting a preference for A/U bases in codons, particularly at the third position. The ENC for the chloroplast genomes of the 15 *Diospyros* species ranged from 45.13 to 45.43 (Table 2), implying weak codon usage bias. The *ycf3* gene displayed the highest ENC value, while the *rps18* gene exhibited the lowest ENC value, except the *rps16* gene with the lowest ENC value in *D. kaki* and *D. vaccinioides* (Table 2).

Analysis of codon composition parameters across all species revealed consistent patterns (Figure 5). GC_all exhibited a significant positive correlation with GC1, GC2, and GC3, with GC1 strongly correlated with GC2. Conversely, correlations between GC1 and GC3, and between GC2 and GC3, were not significant. These findings suggested that base composition at the first and second positions is similar, while the third position differs significantly in GC content. ENC displayed a correlation with GC3 across all species. Significant correlations between ENC and GC1 were observed only in *D. rhombifolia* and *D. cathayensis*, while a significant correlation between ENC and GC2 was noted solely in *D. maulcurei*. No significant correlations were found between ENC and GC_all or between ENC and the number of codons. These results indicate that base composition influences codon preference, particularly at the third position, while the number of codons does not significantly affect ENC.

In the chloroplast genomes of 15 *Diospyros* species, RSCU values of 30 codons exceeded 1 (Appendix A). Notably, 29 of these codons predominantly ended with A or U. Noteworthy preferences included leucine favoring the UUA codon and alanine showing a preference for GCU. Collectively, these chloroplast genomes demonstrated a high degree of uniformity in codon usage, particularly emphasizing a predilection for A or U in the third codon position.

#### 3.3.2. The Causes of Codon Usage Bias

Neutral-plot demonstrated that chloroplast genes from 15 *Diospyros* species were consistently positioned above the diagonal line, with most genes notably deviating from it (Appendix A). The regression coefficients for each species were uniformly modest, ranging from 0.0502 to 0.1326. Notably, the lowest coefficient was observed in *D. sutchunensis*, while the highest was in *D. vaccinioides*, suggesting that mutation pressure explains only 5.02% to 13.26% of the variance, with natural selection accounting for 86.74% to 94.98%.

The analysis of ENC-plot revealed that the GC3s values of individual genes predominantly fall within the range of 0.2 to 0.4, while the corresponding ENC values range from 30 to 55 (Appendix A). With the exception of the *ycf3* gene positioned above the anticipated standard curve, and the *rps2* and *ndhH* genes, which closely followed the curve, the majority of genes exhibited a notable deviation below the standard curve. Notably, a consistent pattern in the frequency distributions across species was illustrated in Appendix A. Approximately 20% of genes fall within the range of −0.05 to 0.05, while the remaining extended beyond this interval and predominantly clustered between 0.05 and 0.15. Collectively, the ENC analysis of *Diospyros* chloroplast genes indicated a weak association between the codon usage biases and GC3s variations, with the codon preferences of the majority of genes primarily shaped by natural selection.

PR2-plot was performed on the A/T and C/G bases at the third position of codons within the chloroplast genomes of 15 *Diospyros* plants species (Appendix A). The analysis revealed an uneven distribution of scatter points across various regions, with a predominant concentration in the lower right quadrant. This distribution suggested an imbalance in the utilization of the four bases at the third codon position in the chloroplast genomes of these *Diospyros* plants, primarily influenced by natural selection. These findings aligned with those obtained from neutral-plot and ENC-plot analyses.

#### 3.3.3. Optimal Codons of Chloroplast Genome in Diospyros

Optimal codon analysis was performed utilizing ENC and RSCU values from the chloroplast genomes of various *Diospyros* species (Figure 6). The range of optimal codons varied from 14 to 21, with *D. oleifera*, *D. tsangii*, and *D. lotus* displaying the highest number at 21. Conversely, *D. hainanensis*, *D. rhombifolia*, *D. cathayensis*, and *D. sutchuensis* exhibited the lowest and identical count of optimal codons. Nine optimal codons were consistently identified across all 15 species, including UGU (cysteine), CAA (glutamine), GAA (glutamic acid), GGU (glycine), CUU and UUA (leucine), AAA (lysine), and GUA and GUU (valine). Notably, only one codon, UUG (leucine), terminated with G, observed solely in *D. vaccinioides*, while the remaining codons concluded with A or U. These findings suggest a pronounced preference for codons ending in A/U in the codon utilization patterns of *Diospyros* chloroplast genomes.

## 4. Discussion

### 4.1. Chloroplast Genome Evolution Within Diospyros

Chloroplast genomes are commonly utilized in studies focusing on genomic evolution, nucleotide substitution patterns, and phylogenetic relationships in plants [57]. This research newly sequenced and annotated the chloroplast genome of *D. tsangii*. The complete chloroplast genome was 157,445 bp in length with 37% GC content (Appendix A), exhibiting a typical quadripartite structure. This species contained 132 coding genes, including 87 CDSs, 8 rRNAs, and 37 tRNAs (Table 1), which is similar to other *Diospyros* species [58,59,60]. The quadripartite boundaries of the chloroplast genomes of 15 *Diospyros* species displayed relative conservation, with the LSC/IRb and SSC/IRa boundaries located within the *rps19* and *ycf1* genes, respectively (Figure 2). Analysis of SSRs revealed 48 to 77 loci in the chloroplast genomes of 15 *Diospyros* species (Figure 1A, Appendix A), predominantly comprising mononucleotide repeats, notably A/T bases (Figure 1B, Appendix A). Furthermore, four types of long-repeat sequences were identified in *D. dumetorum* and *D. maclurei*, while three types were detected in other species, primarily forward repeats and palindromic repeats, potentially involved in chloroplast genome replication and repair mechanisms (Figure 1C, Appendix A). Correlation analysis indicated a significant positive relationship between the length of IR region and the total chloroplast genome length, while no significant correlation was found between the number of SSRs and the total genome length (Figure 1D,E). These findings indicated that the chloroplast genomes of *Diospyros* are generally structurally conserved yet exhibit the genetic diversity within the genus.

Additionally, six highly variable regions were identified, including four intergenic spacers (*trnT*-*trnL*, *petA*-*psbJ*, *psbE*-*petL*, *rpl32*-*trnL*) and two gene regions (*ycf1*, *ndhA*) (Figure 3). In contrast, Li et al. (2018) reported eight highly variable regions (*trnH*-*psbA*, *rps16*-*trnQ*, *rpoB*-*trnC*, *rps4*-*trnT*-*trnL*, *ndhF*, *ndhF*-*rpl32*-*trnL*, *ycf1a*, and *ycf1b*), with some differences possibly attributed to varying sample selections [59]. Considering the overlapping genes identified in the two studies, *trnT*-*trnL*, *rpl32*-*trnL*, *ycf1a*, and *ycf1b* are strongly suggested as potential highly variable region candidates for *Diospyros*, which provided a valuable molecular basis for species identification in *Diospyros*. which may be used as species identification and phylogenetic analysis. Phylogenetic relationships among *Diospyros* have been well resolved (Figure 4), which has also confirmed the strong power of chloroplast genomes in phylogeny. Furthermore, the close phylogenetic relationship among *D. tsangii*, *D. oleifera*, *D. vaccinioides*, and the cultivated persimmon (*D. kaki*) suggested its potential as a promising wild germplasm resource for the breeding of new cultivars.

### 4.2. Natural Selection and the Codon Preference of the Diospyros Chloroplast Genome

Variations in codon usage frequencies among plant genes are a significant feature in plant evolution [49]. Analysis of codon usage bias is a valuable tool for understanding the evolutionary history of plants [23,61,62]. Mutation and natural selection minimally impact the third-position bases of codons, leading to the utilization of the GC3 content for codon analysis [63]. In our study, we observed a descending gradient distribution of GC1 > GC2 > GC_all > GC3 in the chloroplast genomes of 15 *Diospyros* species, with GC3 values ranging from 27.70% to 27.91% (Table 1), indicating a notable preference for A/U bases at the third position of codons. This preference was similar to the findings from chloroplast genome analyses of *Camellia* (GC3 = 28.59–28.64%) [64], Rosaceae (GC3 = 28.27–28.61%) [65], and Juglandaceae (GC3 = 28.2–29.26%) [66], supporting the notion that angiosperm plants tend to favor codons ending with A/U [67,68]. Furthermore, ENC values for the chloroplast genomes of the 15 *Diospyros* species ranged from 45.13 to 45.43 (Table 2). Apart from *rps18*, *rps14*, and *rpl16*, most genes in the 15 species exhibited ENC values above 35, indicating weak codon usage bias and similar bias patterns. This observation aligned with Sharp et al.’s (1988) proposition that species with close genetic relationships exhibit highly similar codon usage biases [69].

Natural selection and mutation are the primary determinants of plant codon preference. Optimal codons are favored by plants due to natural selection, while non-preferred codons can arise from mutations [62,66,70,71]. This study investigated the factors influencing codon usage bias in chloroplast genomes of *Diospyros* plants using ENC-plot, PR2-plot, and neutral-plot analyses. Neutral-plot analysis revealed a weak correlation between GC12 and GC3. Regression analysis indicated that mutational pressure contributed between 5.02% and 13.26% to codon evolution in 15 *Diospyros* species (Appendix A), highlighting the predominant role of natural selection in chloroplast genome codon evolution within this genus. This finding was corroborated by ENC-plot and PR2-plot analyses. ENC-plot analysis showed that most genes deviated below the expected curve (Appendix A), while PR2-plot analysis demonstrated higher usage frequencies of G and C compared to A and T (Appendix A). Collectively, these results indicated that natural selection primarily governs chloroplast codon usage in 15 *Diospyros* species. Moreover, the divergent time estimated showed that the target species *D. tangii* and its sister species *D. oleifera* diverged during the middle Pleistocene (1.62 Ma, 95% HPD: 0.18–5), and the most economical value species *D. kaki* diverged from its closely related species during the late Pleistocene (0.44 Ma, 95% HPD: 0.03–2.57) (Figure 4), which also implied that these species were more susceptible to natural selection than mutation.

Additionally, synonymous and non-synonymous substitution patterns serve as important indicators in the study of gene evolution [72]. Under purifying selection, non-synonymous mutations tend to be eliminated more efficiently than synonymous ones, leading to a lower substitution rate for non-synonymous sites. Consequently, the *dN*/*dS* ratio typically remains below 1 in most instances [73]. To better understand the adaptive evolutionary dynamics of plastomes within the *Diospyros* family, we computed the *dN*/*dS* values for protein-coding genes. Our analysis revealed that only the *psaI* gene exhibited a *dN*/*dS* ratio exceeding 1, while the remaining 79 genes showed ratios less than 1, reflecting widespread and strong purifying selection across the plastid genome (Appendix A). These findings aligned with previous research, such as *Engelhardia* [21], *Camellia* [64], and Theaceae [74]. For instance, the neutrality plot analysis of 13 *Camellia* species conducted by Chen et al. (2023), which reported a mutational contribution rate ranging from 7.68% to 10.02% based on the slope of GC12 to GC3 in this genus [64]. Moreover, in the chloroplast genomes of Theaceae, the slope of GC12 to GC3 is −11.12% to 14.84%, highlighting that natural selection is the primary influence of codon usage bias [74]. Within this context, *Diospyros* plants have developed a codon usage pattern that combines universal elements with taxon-specific features. These results not only lay the groundwork for codon optimization in the molecular breeding of *Diospyros* plants but also shed light on the adaptive strategies adopted by chloroplast genomes in woody plants over extended periods of evolution.

## 5. Conclusions

The genus *Diospyros*, recognized as an economically important group of woody plants, has attracted considerable scientific interest. In this study, a detailed comparative analysis was carried out on the chloroplast genomes of 15 species within the genus, uncovering substantial conservation in GC content, gene content, and overall genomic architecture. A total of 48–77 SSR loci were identified, highlighting their potential application as molecular markers in genetic studies. Codon usage analysis revealed a slight preference for A/U-ending codons in *Diospyros*, a pattern largely attributed to natural selection rather than mutational bias. These results provide critical genomic resources that can support future investigations into molecular evolution and phylogenetic relationships in the genus *Diospyros*.

## Figures and Tables

**Figure 1 biology-14-01568-f001:**
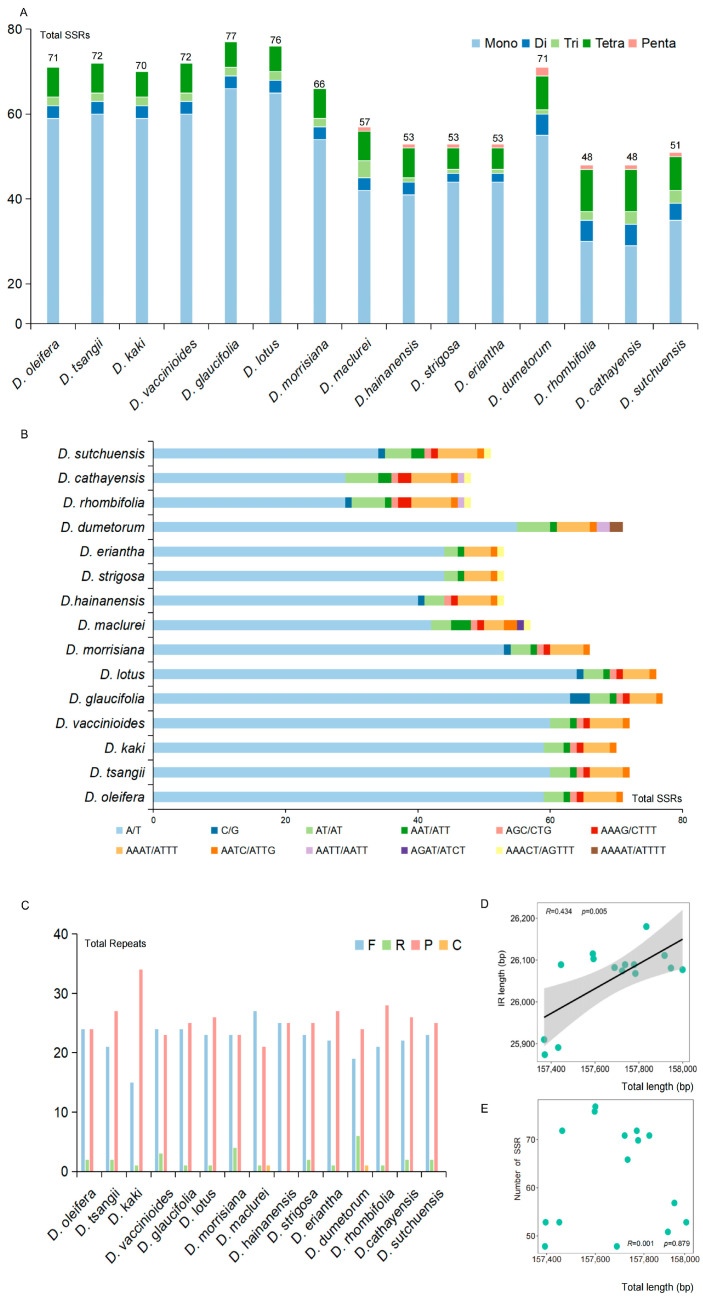
Type and number of SSR in chloroplast genomes of 15 *Diospyros* species. (**A**) The total numbers of SSRs in 15 *Diospyros* species. (**B**) The numbers of different SSRs types in 15 *Diospyros* species. (**C**) The numbers of the repeat sequences in 15 *Diospyros* species. (F: Forward repeat; R: Reverse repeat; P: Palindromic repeat; C: Complement repeat). (**D**,**E**) Analysis of the correlation between genome length and IR length/SSR numbers.

**Figure 2 biology-14-01568-f002:**
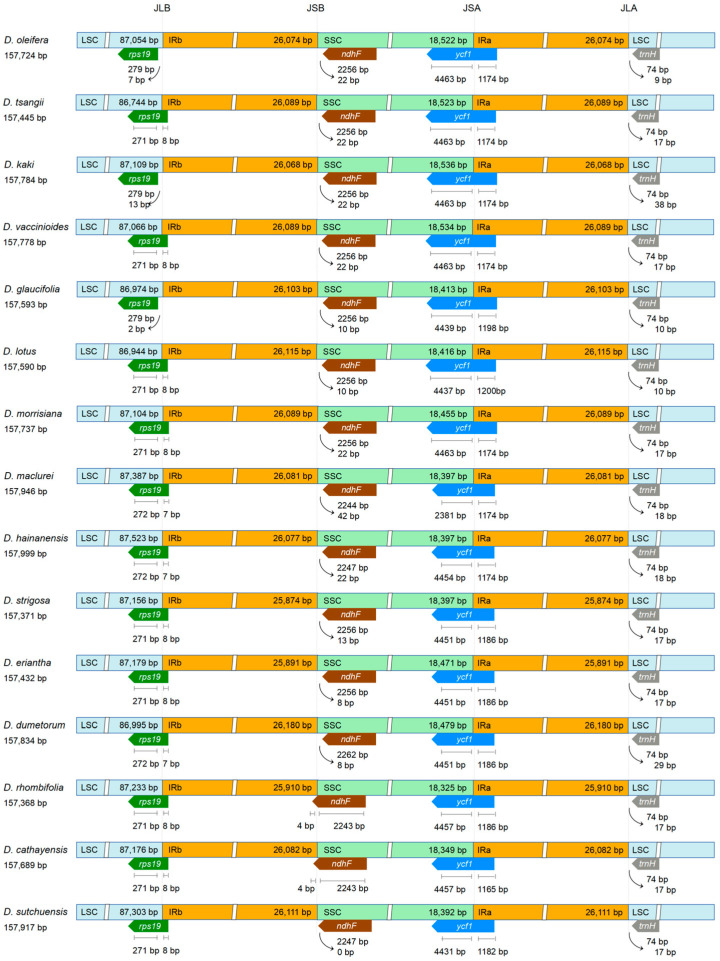
Comparison of LSC, SSC, and IR boundaries revealed both conserved and variable features among the 15 *Diospyros* chloroplast genomes. JLB, JSB, JSA, and JLA denote connection points of adjacent regions. The arrow marks the distance from the gene to the boundary.

**Figure 3 biology-14-01568-f003:**
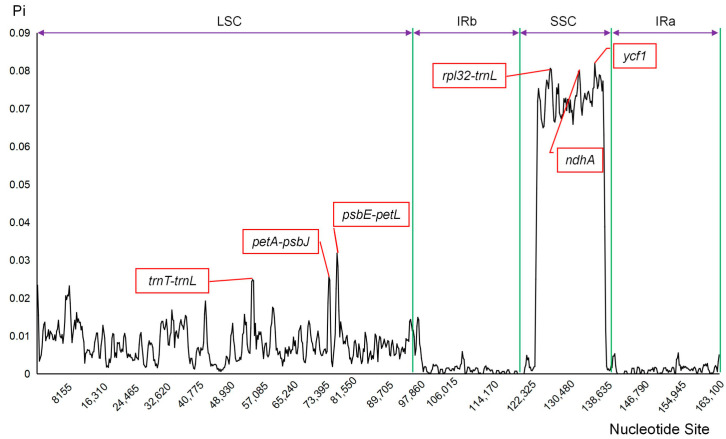
Nucleotide diversity (Pi) analysis of 15 *Diospyros* chloroplast genomes. Top six hypervariable regions of the two datasets were annotated, respectively.

**Figure 4 biology-14-01568-f004:**
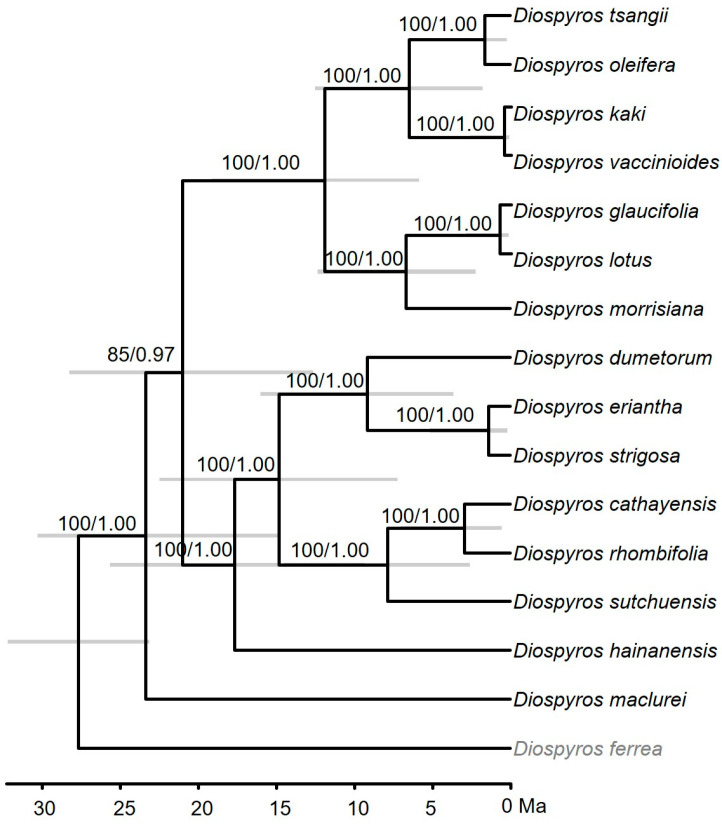
A dating tree based on the complete chloroplast genomes sequences of 15 *Diospyros* species. The numbers on branch represented the supporting values of the Bayesian inference (BI) and maximum parsimony (MP) methods. The following sequences were used: *D. cathayensis*, MF288576; *D. dumetorum*, MF179487; *D. eriantha*, NC_081462; *D. ferrea*, MG049698; *D. glaucifolia*, NC_030784; *D. hainanensis*, NC_042160; *D. kaki*, NC_030789; *D. lotus*, NC_030786; *D. maclurei*, NC_042161; *D. morrisiana*, NC_081461; *D. oleifera*, NC_030787; *D. rhombifolia*, NC_039556; *D. strigosa*, OP480009; *D. sutchuensis*, NC_067511; *D. tsangii*, PX413321; *D. vaccinioides*, NC_060861. The outgroup was marked in light gray.

**Figure 5 biology-14-01568-f005:**
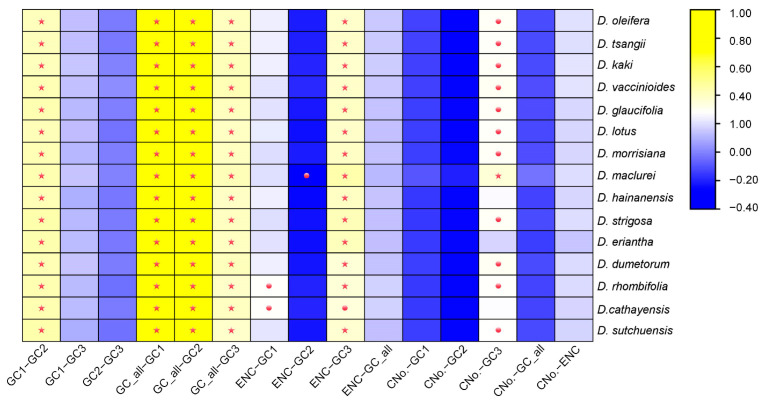
Pearson correlation analysis of GC content, ENC, and codon number across 15 species within the genus *Diospyros*. Asterisks denote statistical significance at *p* < 0.01, while dots indicate significance at *p* < 0.05.

**Figure 6 biology-14-01568-f006:**
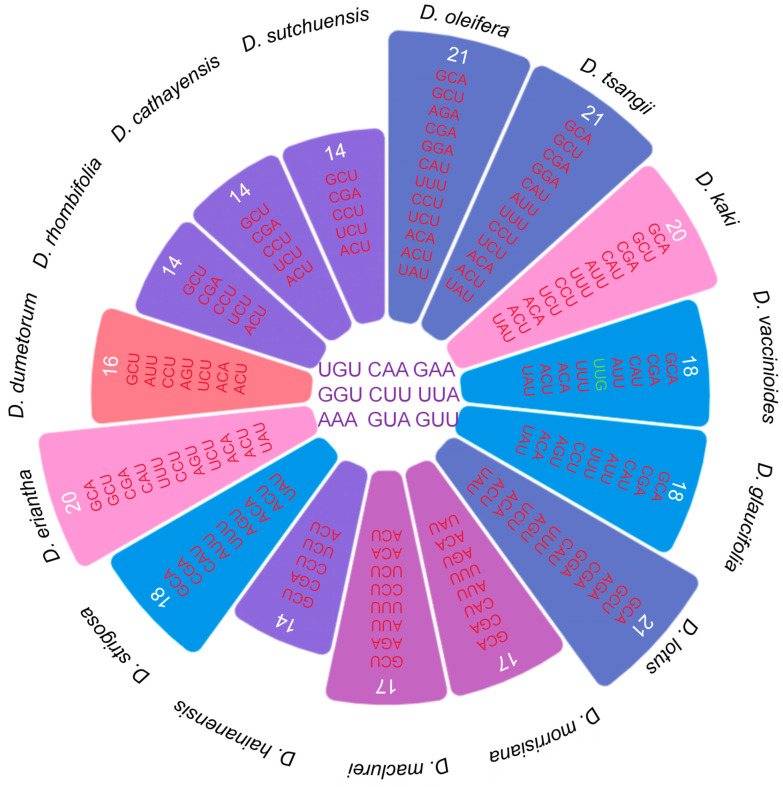
The optimal codons in 15 species chloroplast genomes within the genus *Diospyros*. The 9 codons within the middle circle are the optimal codons shared by 15 species. Green represents codons ending with G or C, while red and purple represent codons ending with A or U.

**Table 1 biology-14-01568-t001:** The complete chloroplast genome features of 15 *Diospyros* species.

Species	ID No.	Genome Size (bp)	LSC Length (bp)	SSC Length (bp)	IR Length (bp)	Gene Content	PCGs	tRNA Genes	rRNA Genes	GC%
*D. oleifera*	NC030787	157,724	87,054	18,522	26,074	132	87	37	8	37.4
*D. tsangii*	PX413321	157,445	86,744	18,523	26,089	132	87	37	8	37.4
*D. kaki*	NC030789	157,784	87,109	18,536	26,068	132	87	37	8	37.4
*D. vaccinioides*	NC060861	157,778	87,066	18,534	26,089	132	87	37	8	37.4
*D. glaucifolia*	NC030784	157,593	86,974	18,413	26,103	132	87	37	8	37.4
*D. lotus*	NC030786	157,590	86,944	18,416	26,115	132	87	37	8	37.4
*D. morrisiana*	NC081461	157,737	87,104	18,455	26,089	132	87	37	8	37.4
*D. maclurei*	NC042161	157,946	87,387	18,397	26,081	132	87	37	8	37.4
*D. hainanensis*	NC042160	157,999	87,523	18,322	26,077	132	87	37	8	37.4
*D. strigosa*	OP480009	157,371	87,156	18,467	25,874	132	87	37	8	37.4
*D. eriantha*	NC081462	157,432	87,181	18,471	25,890	132	87	37	8	37.4
*D. dumetorum*	MF179487	157,834	86,995	18,479	26,180	132	87	37	8	37.4
*D. rhombifolia*	NC039556	157,368	87,233	18,325	25,910	132	87	37	8	37.4
*D. cathayensis*	MF288576	157,689	87,176	18,349	26,082	132	87	37	8	37.4
*D. sutchuensis*	NC067511	157,917	87,303	18,392	26,111	132	87	37	8	37.4

**Table 2 biology-14-01568-t002:** Basic parameters of codon usage bias of chloroplast genome in *Diospyros*.

Species	Codon No.	GC1	GC2	GC3	GC_all	GC3_s_	ENC_AVG_	ENC_MIN_−ENC_MAX_	Genes with ENC ≤ 35
*D. oleifera*	20,911	0.4697	0.3958	0.2788	0.3815	0.2801	45.25	33.81 (*rps18*)−54.25 (*ycf3*)	*rps18*, *rpl16*
*D. tsangii*	20,911	0.4698	0.3958	0.2791	0.3816	0.2805	45.27	33.81 (*rps18*)−54.70 (*ycf3*)	*rps18*, *rpl16*
*D. kaki*	20,913	0.4699	0.3958	0.2788	0.3815	0.2799	45.19	33.75 (*rpl16*)−54.25 (*ycf3*)	*rpl16*, *rps18*
*D. vaccinioides*	20,921	0.4697	0.3950	0.2778	0.3808	0.2789	45.17	33.75 (*rpl16*)−54.25 (*ycf3*)	*rpl16*, *rps18*
*D. glaucifolia*	20,921	0.4695	0.3946	0.2780	0.3807	0.2792	45.38	32.37 (*rps18*)−54.69 (*ycf3*)	*rps18*, *rpl16*
*D. lotus*	20,913	0.4697	0.3954	0.2790	0.3814	0.2803	45.43	32.37 (*rps18*)−54.69 (*ycf3*)	*rps18*, *rpl16*
*D. morrisiana*	20,921	0.4700	0.3944	0.2770	0.3805	0.2782	45.31	33.81 (*rps18*)−54.89 (*ycf3*)	*rps18*, *rpl16*
*D. maclurei*	20,219	0.4697	0.3958	0.2785	0.3813	0.2796	45.38	33.40 (*rps18*)−54.25 (*ycf3*)	*rps18*, *rpl16*
*D. hainanensis*	20,857	0.4709	0.3958	0.2787	0.3818	0.2800	45.31	34.24 (*rpl16*)−53.75 (*ycf3*)	*rpl16*, *rps18*, *rps14*
*D. strigosa*	20,957	0.4705	0.3949	0.2773	0.3809	0.2784	45.13	33.81 (*rps18*)−54.25 (*ycf3*)	*rps18*, *rps14*, *rpl16*
*D. eriantha*	20,958	0.4705	0.3948	0.2775	0.3809	0.2786	45.18	33.81 (*rps18*)−54.25 (*ycf3*)	*rps18*, *rps14*, *rpl16*
*D. dumetorum*	20,921	0.4703	0.3953	0.2788	0.3814	0.2802	45.36	34.47 (*rps18*)−54.25 (*ycf3*)	*rps18*, *rps14*, *rpl16*
*D. rhombifolia*	20,912	0.4706	0.3957	0.2784	0.3816	0.2796	45.21	33.81 (*rps18*)−53.66 (*ycf3*)	*rps18*
*D. cathayensis*	20,905	0.4706	0.3957	0.2790	0.3818	0.2802	45.31	33.81 (*rps18*)−53.66 (*ycf3*)	*rps18*
*D. sutchuensis*	20,886	0.4706	0.3954	0.2785	0.3815	0.2797	45.26	33.81 (*rps18*)−53.66 (*ycf3*)	*rps18*

## Data Availability

The chloroplast genome data of *D. tsangii* have been deposited in the GenBank international repository. The complete list of Accession Numbers corresponding to the samples analyzed in this study is provided within Table 1.

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
