# Peer review of "Evolutionary Dynamics of Chloroplast Genome and Codon Usage in the Genus Diospyros (Ebenaceae)"

_biology, 2025, doi:10.3390/biology14111568_

Round 1
Reviewer 1 Report
Comments and Suggestions for Authors
This manuscript explored the evolutionary dynamics of the chloroplast genome in the Diospyros genus and analyzed the codon usage of the chloroplast genome. The relevant results provided some theoretical support for elucidating phylogenetic relationships and evaluating genetic diversity within the Diospyros genus. However, there are some issues as follows:
- The abstract section does not effectively highlight the key findings or significant results, such as the characteristics of the chloroplastgenome of D. tsangii.
- In the Introduction, the current situation is not favorable reflects the progress of chloroplast genome research due to the lack of novel references.
- This manuscript actually has two aims. It is suggested that (2) and (3) be merged into one.
- In the materials and Methods:
- The materials were obtained from Jianggangshan. However, it is not clearly stated which species these materials are from. Furthermore, the information or data regarding the chloroplast genome of other species are lacking in the manuscript.
- The overall analyses seem simple. Therefore, it is recommended to add the relevant analyses, such as the positive selection analyse. This can help to better explain the pattern of codon usage. Meanwhile, the divergence time of the phylogenetic relationships of the Diospyros genus was also suggested, which can help to explain the phylogenetic relationships among Diospyros species.
- The comparative structure of the chloroplast genome should be extended.
- In the Discussion section, the authors believe that natural selection was a factor for the Diospyros chloroplast genome. However, this was not enough. Thus, it is suggested that the Discussion be mainly reviewed.
- Conclusion section, there are some repeated with the abstract. It should be reviewed.
- It is suggested to place Figure S3 into the main Figure.
- In Figure S3, the numbers on the branch should be annotated.
- The Figures S5 - S8 were missing.
Author Response
Thank you so much for your dedicated effort and thoughtful input on our manuscript. We truly appreciate your time and valuable support.
- The abstract section does not effectively highlight the key findings or significant results, such as the characteristics of the chloroplastgenome of D. tsangii.
Response: Thank you for your valuable comment. We have revised the abstract section as suggested, and specifically added the basic information results of the chloroplast genome of Diospyros tsangii (e.g., genome size, structural characteristics, and other key features) to effectively highlight the core findings of this study (Lines 19-21).
- In the Introduction, the current situation is not favorable reflects the progress of chloroplast genome research due to the lack of novel references.
Response: Thank you for your valuable comment. We have thoroughly revised this section by integrating two recently published (2024–2025) key studies to comprehensively present the current research landscape. (Lines 67-74).
- This manuscript actually has two aims. It is suggested that (2) and (3) be merged into one.
Response: Thank you for your constructive suggestion. We fully agree that merging the two closely related aims enhances the clarity and conciseness of the research objectives. (Lines 90-91).
- In the materials and Methods:
- The materials were obtained from Jianggangshan. However, it is not clearly stated which species these materials are from. Furthermore, the information or data regarding the chloroplast genome of other species are lacking in the manuscript.
Response: Thank you for your constructive suggestion. We have revised as your suggestion (Lines 97-101).
- The overall analyses seem simple. Therefore, it is recommended to add the relevant analyses, such as the positive selection analyse. This can help to better explain the pattern of codon usage. Meanwhile, the divergence time of the phylogenetic relationships of the Diospyros genus was also suggested, which can help to explain the phylogenetic relationships among Diospyros species.
Response: Thank you for your constructive suggestions to enrich the study’s analytical depth. We have carefully addressed your comments and supplemented the relevant analyses in the revised manuscript. We have added positive selection analysis as recommended. This analysis helps better interpret the pattern of codon usage by identifying codons or genes under selection pressure, which enhances the robustness of our conclusions on codon evolution (Lines 132-142). We have supplemented the divergence time estimation for the phylogenetic relationships of the Diospyros genus (Lines 151-159).
- The comparative structure of the chloroplast genome should be extended.
- In the Discussion section, the authors believe that natural selection was a factor for the Diospyros chloroplast genome. However, this was not enough. Thus, it is suggested that the Discussion be mainly reviewed.
Response: Thank you for your valuable suggestion. We have fully addressed this comment in the revised manuscript. (Lines 454-469)
- Conclusion section, there are some repeated with the abstract. It should be reviewed.
Response: We have fully addressed this comment in the revised manuscript. (Lines 482-491)
- It is suggested to place Figure S3 into the main Figure.
Response: Revised as your suggestion (the updated Figure 4).
- In Figure S3, the numbers on the branch should be annotated.
Response: We sincerely appreciate your suggestion. We have fully addressed this comment in the revised manuscript (the updated Figure 4, Lines 305-306)
- The Figures S5 - S8 were missing.
Response: We truly appreciate your careful review and valuable comments. We have supplemented the relevant attached figures according to your suggestions.

Reviewer 2 Report
Comments and Suggestions for Authors
In this work, the authors investigated the chloroplast genome and codon usage across 15 Diospyros species. Overall, all analyzed species display a typical quadripartite genomic structure, along with highly conserved gene composition and GC content. Several variable intergenic and coding regions have been identified, which are considered promising candidates for DNA barcoding. Similar to other species, natural selection is suggested to be the primary force driving codon usage bias in the chloroplast genomes of Diospyros species, shown as a mild preference towards A/U-ending codons. This work provides new insights into chloroplast genomic information of Diospyros species and establishes a theoretical foundation for chloroplast genetic engineering work, especially for the Diospyros plants with significant ecological and economic potential.
Comments:
- There is a typo in line 122: “telated”. Should this be “related” to codon usage bias?
- In the section 2.4.1, the authors should explain the rationale for only using genes exceeding 300bp for codon usage analysis. This leads to 40% of the protein-coding genes are omitted in the analysis. Many of these excluded genes, such as psbE, psbF, and atpH, are essential chloroplast genes with high gene expression level, more importantly, play critical roles in photosynthesis and chloroplast function.
- The authors list six intron-containing tRNA genes, which are also found in the chloroplasts of other model species, except trnG-GCC. In other model species, chloroplast trnG-UCC contain one intron, rather than trnG-GCC. The authors show clarify if this is accurate in their findings.
- The rationale behind the introduction of chloroplast genome information for D. tsangii in section 3.1, followed by the introduction of all 15 Diospyros species in section 3.2 is unclear. This creates significant redundancy between sections 3.1 and 3.2.1. If there is a specific reason for introducing D. tsangii first, the authors should clearly explain this.
- In Figure 1A-C and Figure3, axis labels are missing. For Figure3, an appropriate figure legend is missing.
Author Response
Dear Reviewer,
Thank you so much for your dedicated effort and thoughtful input on our manuscript. We truly appreciate your time and valuable support.
Comments and Suggestions for Authors
In this work, the authors investigated the chloroplast genome and codon usage across 15 Diospyros species. Overall, all analyzed species display a typical quadripartite genomic structure, along with highly conserved gene composition and GC content. Several variable intergenic and coding regions have been identified, which are considered promising candidates for DNA barcoding. Similar to other species, natural selection is suggested to be the primary force driving codon usage bias in the chloroplast genomes of Diospyros species, shown as a mild preference towards A/U-ending codons. This work provides new insights into chloroplast genomic information of Diospyros species and establishes a theoretical foundation for chloroplast genetic engineering work, especially for the Diospyros plants with significant ecological and economic potential.
Comments:
- There is a typo in line 122: “telated”. Should this be “related” to codon usage bias?
Response: We have corrected the error (Line 164).
- In the section 2.4.1, the authors should explain the rationale for only using genes exceeding 300bp for codon usage analysis. This leads to 40% of the protein-coding genes are omitted in the analysis. Many of these excluded genes, such as psbE, psbF, and atpH, are essential chloroplast genes with high gene expression level, more importantly, play critical roles in photosynthesis and chloroplast function.
Response: Thank you very much for your valuable comment. We fully agree with your suggestion and have made corresponding revisions in Section 2.4.1 (Lines 165-171).
- The authors list six intron-containing tRNA genes, which are also found in the chloroplasts of other model species, except trnG-GCC. In other model species, chloroplast trnG-UCCcontain one intron, rather than trnG-GCC. The authors show clarify if this is accurate in their findings.
Response: Thank you very much for your comment. We have carefully verified the intron distribution of chloroplast trnGUCC genes in model species and made corresponding revisions to the manuscript (Line 224).
- The rationale behind the introduction of chloroplast genome information for tsangiiin section 3.1, followed by the introduction of all 15 Diospyros species in section 3.2 is unclear. This creates significant redundancy between sections 3.1 and 3.2.1. If there is a specific reason for introducing D. tsangii first, the authors should clearly explain this.
Response: Thank you very much for your constructive comment. We fully understand your concern about the logical connection and potential redundancy between Sections 3.1 and 3.2.1, and have made targeted revisions to clarify the structure.
The original arrangement aimed to distinguish two core focuses: Section 3.1 specifically presents the chloroplast genome characteristics of D. tsangii—the newly sequenced species in this study—serving as the foundational data for subsequent comparative and evolutionary analyses. Section 3.2.1, by contrast, expands to 15 Diospyros species to discuss plastid genome evolution from a genus-wide perspective. To eliminate ambiguity and redundancy, we have supplemented explicit introductory sentences in Section 3.1 to emphasize that D. tsangii is the novel sequencing data of this study, and added logical transition statements between the two sections to clarify their hierarchical relationship (i.e., "based on the newly obtained D. tsangii chloroplast genome, we further conducted comparative evolutionary analysis with 14 other Diospyros species"). Additionally, we have streamlined overlapping descriptive content to enhance readability.
- In Figure 1A-C and Figure3, axis labels are missing. For Figure3, an appropriate figure legend is missing.
Response: We have revised as your comment (Lines 250-254, 280-281).

Reviewer 3 Report
Comments and Suggestions for Authors
This manuscript presents a well-structured and clearly written study on Diospyros plastomes. The paper effectively communicates the significance and potential applications of this work in plant biology and genomics. Overall, the manuscript is suitable for publication in Biology after the suggested revisions.
The abstract provides a clear summary of the study’s purpose and findings, highlighting the promise and importance of the research. The author may consider mentioning the newly sequenced plastome of D. tsangii to make the contribution of this study more explicit.
The introduction is well-organised and logically presented. It effectively summarises the biological importance of Diospyros and the utility of chloroplast genomes. The rationale behind the analyses is clearly explained, with only a few minor typographical or stylistic issues noted.
The materials and methods section is generally clear and well written. Only minor edits are suggested to improve clarity and readability.
The results are clearly presented and mostly well reported. However, a few ambiguous points require clarification. In particular, the phylogenetic tree is unrooted but displayed as if it were rooted. The authors should either clearly state that the tree is unrooted or modify its presentation to an appropriate unrooted format.
The discussion successfully brings the findings to a broader framework of plant biology and highlights the potential applications of Diospyros plastome data.
Please refer to the table below for detailed comments and suggested corrections.
|
Section |
Line |
Error / Notice |
Correction |
Comments / Suggestion |
|
Introduction |
28 |
Linn. |
L. |
|
|
Introduction |
32 |
This genus [is the economically…] |
Diospyros [is the economically…] |
To avoid repetition of "this genus" |
|
Introduction |
41 |
The chloroplast genomes [have been proven…] |
Either: Chloroplast genomes [have been proven..] OR The chloroplast genome [has been proven…] |
|
|
Material and methods |
75 |
The plant materials were obtained… |
|
The author should provide slightly more details on plant materials (e.g., how many samples / species were collected, how were they preserved) |
|
Material and methods |
83 |
low-quality region |
|
What was your quality score cut-off? Probably easier to provide part of the command as this will be useful for readers (e.g., LEADING:3 TRAILING:3 SLIDINGWINDOW:4:20 MINLEN:50) |
|
Material and methods |
93 |
15 Diospyros speces |
|
Why these 15 species when there are plastomes of other species available as well? The authors should provide some rationale why they chose these species. |
|
Material and methods |
113 |
Phylogenetic tree reconstruction |
|
There is no statement of outgroup. It is okay to use an unrooted tree but you should state it clearly and display an unrooted tree. The tree shown in Figure S3 looks like a rooted tree. |
|
Material and methods |
161 |
calculated as [RSCU…] |
|
Wrong font or size |
|
Results |
202 |
…, while no significant correlation was found between the number of SSRs… |
|
What about the total length of SSRs from each plastome? Does it have any correlation with the plastome size? The authors don’t need to do additional analysis but in case it’s already been done, it might be worth reporting it here. |
|
Results |
207 |
F: Forward repeat; R: Reverse repeat.. |
(F: Forward repeat; R: Reverse repeat..) |
These legend descriptions should be put inside a bracket to give the figure description better readability. |
|
Results |
217 |
ndhH |
ndhF |
|
|
Results |
242 |
The remaining Diospyros species formed a clade with well supporting values. |
|
It is true that the rest of the species fall into the same clade, but I would argue that D. hainanensisand D. maclurei are closely related to the rest of the species in this clade. You may say something like: the clade containing the rest of the species has D. hainanensis and D. maclurei as successive sister clades while the rest of the species are form a less divergent cluster. |
|
Results |
306 |
Diospyros (non-italic) |
Diospyros (italic) |
Make Diospyros italic or different from the normal text |
|
Discussion |
323 / 356 |
Diospyros (italic) |
Diospyros (non-italic) |
If the title is in italic, the genus name should not be italicised |
|
Discussion |
341 |
These findings indicated that the chloroplast genomes of Diospyros are generally highly conserved. |
(Optional) These findings indicated that the chloroplast genomes of Diospyros are generally structurallyconserved. |
Probably consider adding “structurally” conserved as this will show better contrast with your next sentence with somewhat moderate nucleotide (genetic) diversity. |
|
Discussion |
347-350 |
Therefore, trnT-trnL … which may be used as species identification and phylogenetic analysis. |
|
Convolute sentence. Consider rephrase the sentence. |
|
References |
430 |
Forst, J.R.; Forst, G. |
S. K. Lee, M.G. Gilberg & F. White |
Wrong authors |
|
Supplementary information |
|
Table S1. List of gene contents in Diospyros tsangii. |
Table S1. List of gene contents in the plastome of Diospyros tsangii. |
|
|
Supplementary information |
|
Table S2. Types and amounts of SSRs within the chloroplast genomes of fifteen Diospyros species. |
Table S2. Types and numbers of SSRs within the chloroplast genomes of fifteen Diospyros species. |
|
|
|
|
Table S3. Types and amounts of the repeat sequences in fifteen Diospyros species. |
Table S3. Types and numbers of the repeat sequences in fifteen Diospyros species. |
|
Author Response
Dear Reviewer,
Thank you so much for your dedicated effort and thoughtful input on our manuscript. We truly appreciate your time and valuable support.
Comments and Suggestions for Authors
This manuscript presents a well-structured and clearly written study on Diospyros plastomes. The paper effectively communicates the significance and potential applications of this work in plant biology and genomics. Overall, the manuscript is suitable for publication in Biology after the suggested revisions.
The abstract provides a clear summary of the study’s purpose and findings, highlighting the promise and importance of the research. The author may consider mentioning the newly sequenced plastome of D. tsangii to make the contribution of this study more explicit.
Response: Thanks very much for your effort on our manuscript. We have revised as your suggestion (Lines 19-22).
The introduction is well-organised and logically presented. It effectively summarises the biological importance of Diospyros and the utility of chloroplast genomes. The rationale behind the analyses is clearly explained, with only a few minor typographical or stylistic issues noted.
Response: Thanks very much for your comment. We have revised as your suggestion through the Introduction.
The materials and methods section is generally clear and well written. Only minor edits are suggested to improve clarity and readability.
Response: Thanks very much for your comment. We have revised the materials and methods section to make it clearer and more readable.
The results are clearly presented and mostly well reported. However, a few ambiguous points require clarification. In particular, the phylogenetic tree is unrooted but displayed as if it were rooted. The authors should either clearly state that the tree is unrooted or modify its presentation to an appropriate unrooted format.
Response: Thanks very much for your comment. We have revised the phylogenetic tree (the new Figure 4).
The discussion successfully brings the findings to a broader framework of plant biology and highlights the potential applications of Diospyros plastome data.
Response: Thanks very much for your effort on our manuscript.
Please refer to the table below for detailed comments and suggested corrections.
|
Section |
Line |
Error / Notice |
Correction |
Comments /Suggestion |
Response |
|
Introduction |
28 |
Linn. |
L. |
|
We truly appreciate your careful review and valuable comments. We have corrected the error in line 42. |
|
Introduction |
32 |
This genus [is the economically…] |
Diospyros [is the economically…] |
To avoid repetition of "this genus" |
Thank you for your valuable feedback—we have carefully incorporated your suggestions into the revisions. (line 48) |
|
Introduction |
41 |
The chloroplast genomes [have been proven…] |
Either: Chloroplast genomes [have been proven..] OR The chloroplast genome [has been proven…] |
|
Thank you for your valuable feedback—we have carefully incorporated your suggestions into the revisions. (line 56) |
|
Material and methods |
75 |
The plant materials were obtained… |
|
The author should provide slightly more details on plant materials (e.g., how many samples / species were collected, how were they preserved) |
Thank you for your valuable suggestion. We have fully addressed this comment in the revised manuscript. (line 99) |
|
Material and methods |
83 |
low-quality region |
|
What was your quality score cut-off? Probably easier to provide part of the command as this will be useful for readers (e.g., LEADING:3 TRAILING:3 SLIDINGWINDOW:4:20 MINLEN:50) |
Thank you for your valuable suggestion. We have fully addressed this comment in the revised manuscript. (line 108) |
|
Material and methods |
93 |
15 Diospyros speces |
|
Why these 15 species when there are plastomes of other species available as well? The authors should provide some rationale why they chose these species. |
We sincerely appreciate your suggestion and have carefully included the corresponding explanations in the manuscript. (line 120) |
|
Material and methods |
113 |
Phylogenetic tree reconstruction |
|
There is no statement of outgroup. It is okay to use an unrooted tree but you should state it clearly and display an unrooted tree. The tree shown in Figure S3 looks like a rooted tree. |
We sincerely appreciate your suggestion. We have fully addressed this comment in the revised manuscript. (line 161) |
|
Material and methods |
161 |
calculated as [RSCU…] |
|
Wrong font or size |
|
|
Results |
202 |
…, while no significant correlation was found between the number of SSRs… |
|
What about the total length of SSRs from each plastome? Does it have any correlation with the plastome size? The authors don’t need to do additional analysis but in case it’s already been done, it might be worth reporting it here. |
We highly appreciate this valuable perspective, as it provides a meaningful direction for further exploring plastome characteristics. Due to the focus of our current study on genetic diversity analysis, we have not conducted additional analysis on this specific correlation and thus do not have relevant data to report at present.We agree that this correlation is worthy of attention, and we will consider incorporating this analysis in our future follow-up studies to enrich the research findings. |
|
Results |
207 |
F: Forward repeat; R: Reverse repeat.. |
(F: Forward repeat; R: Reverse repeat..) |
These legend descriptions should be put inside a bracket to give the figure description better readability. |
Thank you for your valuable feedback—we have carefully incorporated your suggestions into the revisions. (line 256) |
|
Results |
217 |
ndhH |
ndhF |
|
We truly appreciate your careful review and valuable comments. We have corrected the error in line 266. |
|
Results |
242 |
The remaining Diospyros species formed a clade with well supporting values. |
|
It is true that the rest of the species fall into the same clade, but I would argue that D. hainanensisand D. maclurei are closely related to the rest of the species in this clade. You may say something like: the clade containing the rest of the species has D. hainanensis and D. maclurei as successive sister clades while the rest of the species are form a less divergent cluster. |
Thank you so much for your thoughtful suggestions. We truly appreciate your input and have carefully incorporated the changes based on your feedback. (lines 300-303) |
|
Results |
306 |
Diospyros (non-italic) |
Diospyros (italic) |
Make Diospyros italic or different from the normal text |
Thank you so much for your helpful suggestions. We’ve carefully reviewed the entire manuscript and have updated the formatting of "Diospyros" in accordance with your feedback. |
|
Discussion |
323 / 356 |
Diospyros (italic) |
Diospyros (non-italic) |
If the title is in italic, the genus name should not be italicised |
Thank you so much for your helpful suggestions. We’ve carefully reviewed the entire manuscript and have updated the formatting of "Diospyros" in accordance with your feedback. |
|
Discussion |
341 |
These findings indicated that the chloroplast genomes of Diospyros are generally highly conserved. |
(Optional) These findings indicated that the chloroplast genomes of Diospyros are generally structurallyconserved. |
Probably consider adding “structurally” conserved as this will show better contrast with your next sentence with somewhat moderate nucleotide (genetic) diversity. |
Thank you for your valuable feedback—we have carefully incorporated your suggestions into the revisions. (line 416) |
|
Discussion |
347-350 |
Therefore, trnT-trnL … which may be used as species identification and phylogenetic analysis. |
|
Convolute sentence. Consider rephrase the sentence. |
We sincerely appreciate your suggestion and have carefully included the corresponding explanations in the manuscript. (line 421) |
|
References |
430 |
Forst, J.R.; Forst, G. |
S. K. Lee, M.G. Gilberg & F. White |
Wrong authors |
We truly appreciate your careful review and valuable comments. We have corrected the error in line 522. |
|
Supplementary information |
|
Table S1. List of gene contents in Diospyros tsangii. |
Table S1. List of gene contents in the plastome of Diospyros tsangii. |
|
Thank you for your valuable feedback—we have carefully incorporated your suggestions into the revisions. |
|
Supplementary information |
|
Table S2. Types and amounts of SSRs within the chloroplast genomes of fifteen Diospyros species. |
Table S2. Types and numbers of SSRs within the chloroplast genomes of fifteen Diospyros species. |
|
Thank you for your valuable feedback—we have carefully incorporated your suggestions into the revisions. |
|
|
|
Table S3. Types and amounts of the repeat sequences in fifteen Diospyros species. |
Table S3. Types and numbers of the repeat sequences in fifteen Diospyros species. |
|
Thank you for your valuable feedback—we have carefully incorporated your suggestions into the revisions. |

Round 2
Reviewer 1 Report
Comments and Suggestions for Authors
All issues were basically solved.
In the Abstract, it is suggested that lines 19 - 22 about the chloroplast genome of D. tsangii be placed before line 25 ‘Our findings...’